# The Role of Palynology in Archaeoecological Research: Reconstructing Human-Environment Interactions during Neolithic in the Western Mediterranean

Jordi Revelles [1,2]

1    Institut Català de Paleoecologia Humana i Evolució Social (IPHES-CERCA), Zona Educacional 4,
     Campus Sescelades URV (Edifici W3), 43007 Tarragona, Spain; jordi.revelles@gmail.com
2    Departament d'Història i Història de l'Art, Universitat Rovira i Virgili, Avinguda de Catalunya 35,
     43002 Tarragona, Spain

**Abstract:** This paper provides an overview of the potential of palynology within palaeoenvironmental research to reconstruct past landscapes and assess the relationship between vegetation and the first farming communities. The analysis of pollen and non-pollen palynomorphs in natural records evidenced how the adoption of farming and new sedentary settlement patterns resulted in major landscape transformation on extra-local or regional scales in the Western Mediterranean, affecting sclerophyllous and riparian forests in North Corsica, Mediterranean maquis in South Corsica, and oak forests in NE Iberia. In addition, palynology has been confirmed as a relevant source of data to address the local palaeoenvironmental evolution in lakes, wetlands, and archaeological sites, providing evidence of the presence of flocks (spores of coprophilous fungi), and changes in hydrology (salinity, dryness/wetness, aquatic/palustrine phases) and in geomorphology (soil erosion indicators). Finally, the spatial analysis of pollen and NPP intra-site distribution is presented here as a valuable tool to assess the social use of space in archaeological sites. In that sense, archaeopalynology has provided detailed information about site formation processes, social use of space, and the use of plants and fungi in the site of La Draga (Girona, Spain).

**Keywords:** archaeopalynology; archaeoecology; bioarchaeology; Middle Holocene; Iberian Peninsula; La Draga; Corsica

## 1. Introduction

The adoption of the farming lifestyle changed the way in which humans and nature interacted, resulting in the onset of an expanding process of landscape transformation [1,2]. While environmental changes during the Early Holocene were climatically driven, the role of human activities became decisive during the Middle Holocene and increased exponentially from the Late Holocene [3]. Indeed, the complex combination of climatic and anthropogenic factors since ~8.0 cal kyr BP often makes it difficult to distinguish the influence of these two variables on the Mediterranean landscape [4–6]. In any case, the emergence of farming activities resulted in the transformation of landscapes, as recorded in pollen [7–13] and bioarchaeological records [14–17] in the Mediterranean area.

The Mediterranean region is in the focal point of current challenges concerning global warming, sea-level rise and other natural processes such as wildfires, and in this context, long-term archaeological and palaeoecological data provide a reference period against which current conditions are compared in order to understand the dynamics linking climate, vegetation, fire, and human activities [18]. The Western Mediterranean region has great potential, offering a high variability of ecosystems and different settlement dynamics since Prehistory, in which the early Neolithisation of Corsica and Sardinia (~8000 cal BP, [19]), and later in Iberia (~7500 cal BP, [20]) is remarkable. The first farming communities, emerged in the Near East ca. 12 kyr cal BP, rapidly spread in the Eastern Mediterranean between 8.5

and 8.2 kyr cal BP, but climatic and environmental constraints related to the 8.2 kyr cal BP cooling event slowed down this expansion [21], resulting in an arrhythmic spread from Eastern to Western Mediterranean. In that sense, in different regions of the Mediterranean basin, a gap in archaeological evidence exists in the transition from the 9th to the 8th millennium cal BP [22].

Detailed palaeoenvironmental research is required to reconstruct past landscapes and to assess the relationship between vegetation, climate, and the first farming communities. Despite the changes in food production, management of resources and settlement patterns deriving in socioeconomic transformation, archaeological research on Neolithisation must not be limited to this cultural transformation and should be studied from a socio-environmental perspective [22], assessing the role of climate and environmental processes in this historical change on the basis of a geoarchaeological or archaeoecological approach. At this point, archaeoecology is defined as the integration of palaeoecological and archaeological research to study the dialectical relationship between humans and environment, between social organization and the geosystem. In order to reconstruct the environmental framework and possible constraints on human and social evolution, as well as the impacts of anthropogenic work and human-induced disturbance processes, the unit of study must transcend from the archaeological site to the landscape, understood as a historical product transformed by both natural and social processes. This approach is based on the assumption that human societies are not a passive element whose actions are determined by climatic or environmental changes, since social needs are decisive in the management of natural resources, depending on different modes of production and organizational strategies [23]. Therefore, it becomes essential to integrate the palaeoecological study of Mid-Holocene climate fluctuations and landscape evolution with archaeological and bioarchaeological data to assess human-environment interactions during the Neolithic.

The main aim of this paper is to give an overview on the potential of palynology in archaeoecological research on the first farming communities. It provides a synthesis of Mid-Holocene pollen records from two regions in the Western Mediterranean: North-East Iberia and Corsica, to identify human impact imprints in pollen records and ecological changes and disturbance processes linked to human activities. In addition, a review is given on the potential of archaeopalynology to address palaeoenvironmental and archaeological issues, such as the management of vegetal resources and the social use of space, focusing on the case study of the early Neolithic site of La Draga (Banyoles, Girona, Spain).

## 2. Study Area: Environmental Settings and Archaeological Background

### 2.1. Corsica

The climate of Corsica is characterized by a steep altitudinal gradient, from the warm and dry lower elevations (average annual temperature of ~14 to 17 °C) to the cold and humid higher altitudes (9–13 °C). The climate of North Corsica is cooler and wetter, with a mean annual temperature of 15.5 °C and mean annual precipitationof 771mm in Bastia (North Corsica); while the climate of South Corsica iswarmer and drier (16.4 °C and 542mm in Bonifacio).The broad altitudinal range of the island results in several forest zones. The lowest elevationsare characterized by the predominance of sclerophyllous evergreen oak forest (*Quercus ilex*, *Quercus suber*) and Mediterranean scrubland or maquia (*Erica arborea*, *Pistacia lentiscus*, *Cistus monspeliensis* and *Juniperus phoenicea*). At intermediate elevations, mesophyllous pine forest (*Pinus pinaster*) is widespread, and mixed deciduous forest (*Quercus pubescens*, *Quercus petraea*, *Ostrya carpinifolia*, *Alnus cordata*, *Castanea sativa*) is locally abundant. In North Corsica, various types of maquia occur in the surroundings of Saint Florent, including stands of wild olive (*Olea europaea*) and lentisk (*Pistacia lentiscus*) in the warm coastal areas; tall maquis with *Arbutus*; evergreen oak and heather (*Calluna vulgaris*) in the cooler areas; and low maquis dominated by *Cistus*, *Rosmarinus officinalis*, and *Lavandula stoechas* on poor soils (sands) and in rocky environments. In South Corsica, the area of l'étang de Piantarella is occupied by Mediterranean maquis (*Cistus monspeliensis*,

*Juniperus phoenicea*, *Pistacia lentiscus*, *Olea europaea*, *Arbutus unedo*) and marshland in a more open landscape.

The first human settlement in Corsica is marked by a sharp discontinuity between the Mesolithic and Neolithic [19]. The first attested human presence is at the onset of the Holocene, consisting of hunter-gatherer communities that did not establish permanent settlements but rather occupied caves and other shelters. Then, a gap in the population is documented during the transition from the 9th to the 8th millennium cal BP, suggesting that agropastoral colonisation occurred in depopulated areas, as confirmed by genetic discontinuity between these two populations [19]. The first Neolithic evidence is documented at ~7.8 cal kyr BP, followed by the rapid colonisation of the island by the first farming societies. However, there is little evidence of the early development of agriculture in archaeobotanical records in Corsica, strongly supporting, thus, the need for archaeoecological or geoarchaeological research to detect the earliest evidence of agriculture on the island [24]. From the Middle Neolithic (6.8–5.9 cal kyr BP, [25]), human settlements were concentrated in coastal regions and the nearby plains.

### 2.2. NE Iberia

The climate in the NE Iberian Peninsula is defined as Mediterranean, with variations depending on the altitude and distance from the sea. The climate in Pla de l'Estany, the plain around Lake Banyoles (173 m asl.), 35 km from the coast and 50 km south of the Pyrenees, is defined as humid Mediterranean or sub-Mediterranean, characterized by higher precipitation and cooler temperatures than in the coastal ranges, with an annual precipitation of 750 mm and a mean annual temperature of 15 °C in Banyoles. The average maximum temperature during July andAugust is 23 °C, and the average minimum is 7 °C in winter. At the present, the vegetation in the NE Iberian Peninsula is characterized by the predominance of evergreen sclerophyllous forests (evergreen oaks and Mediterranean pines) on the Mediterranean coast and in the continental interior, and humid sub-Mediterranean forests (deciduous oak and beech) in the middle mountains of the pre-Pyrenees. Pla de l'Estany consists of a transition area between humid forests in the nearby mountainous areas (eastern slopes of pre-Pyrenees) and sclerophyllous forests and maquia on the coast. Dense vegetation formations in the mountains surrounding Lake Banyoles, are dominated by a mixed evergreen forest (*Quercus ilex*, *Quercus coccifera*, *Rhamnus alaternus*, *Phillyrea media*, *P. angustifolia*), deciduous oak (*Quercus pubescens*, *Buxus sempervirens*, *Ilex aquifolium*) and pine forest (*Pinus halepensis*). In this context, shrublands (*Erica arborea*, *Rosmarinus officinalis*) are well represented.

The first evidence of Neolithic occupation in NE Iberia is dated in the second half of the 8th millennium cal. BP. The archaeological evidence suggests a chronological hiatus of ~500 years, in ~8.1–7.6 cal kyr BP, in which no human occupations have been documented [26,27]. The gap in human presence in the region during the first half of the 8[th] millennium cal BP indicate that the Neolithisation of the NE Iberian Peninsula was the result of migration of farming populations to uninhabited territories [28], contrasting with the interaction documented between indigenous Mesolithic communities and the arriving farming communities on the Eastern coast of the Iberian Peninsula [29]. Bioarchaeological research carried out in recent decades has shown that intensive mixed farming (small-scale, labour-intensive cultivation integrated with small-scale herding) emerges as the most plausible model in NE Iberia, featuring an agriculture that included the cultivation of a large variety of cereals and legumes [30,31].

Similarities in the archaeological record, particularly the Cardium-impressed ware, between the Italian and the Iberian Peninsula, including south France, Corsica and Sardinia, point to the existence of active maritime networks, which would have played a significant role in the Neolithisation process [32]. As mentioned, both in Corsica and in NE Iberia, the first farming societies reached uninhabited territories after the cooling phase associated with the 8.2 kyr cal BP event, in regions predominated by natural landscapes free of human disturbance. In both areas, Neolithic communities settled along the coast and in wetland

areas, where they found a rich variety of ecological niches for provision of food and raw materials.

### 3. Materials and Methods

*3.1. Study Cases*

3.1.1. Piantarella (South Corsica)

L'étang de Piantarella, is located in Bonifacio, on the southern coast of Corsica (France) (Figure 1). Three cores were extracted: Piantarella 1 (630 cm, 0.36 m asl.; Piantarella 2 (315 cm, 0.33 m a.s.l.); Piantarella 3 (420 cm, 0.19 m a.s.l.). In Piantarella 1, pollen analysis was carried out on samples from the organic basal part (515–610 cm) of the core, dated in 7.4–7.2 kyr cal BP; in Piantarella 2 on samples at 5–69 and 130–145 cm, dated in 4.7–4.6 kyr cal BP and in 0.4 kyr cal BP to present; and in Piantarella 3 on samples at 250–381 cm, dated in 6.6–5.3 kyr cal BP [24,33].

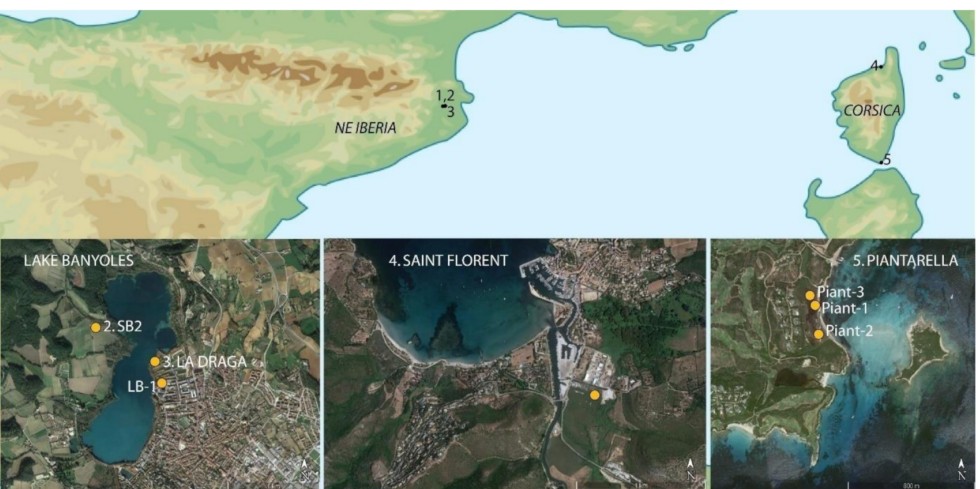

**Figure 1.** Location of study areas, pollen records, and archaeological sites.

3.1.2. Saint Florent (North Corsica)

The studied core (0.47 m asl.) was drilled in the modern swampland of the Aliso River mouth, to the east of the modern harbour, in Saint Florent, on the northern coast of Corsica (France) (Figure 1). Saint Florent consists of a 730 cm long core and pollen analysis included samples from different sections of the core providing data from 6.3 kyr cal BP to the present [24,34].

3.1.3. Lake Banyoles (NE Iberia)

Lake Banyoles (Girona, Spain) (Figure 1) is a karst lake associated with a large karst aquifer system located in a tectonic depression, fed by underground water. The lake isapproximately 2100 m long and 750 m wide with an average depth of 15 m that in several places reaches up to 46 m [35,36]. Two pollen records are dated in Mid-Holocene chronologies. Firstly, the Lake Banyoles core (LB) [37] was obtained from the eastern shore of the lake. This 3310 cm long core provided a continuous pollen record for the last 30,000 years. However, this paper focuses on the first 410 cm, dated in the Early to Middle Holocene transition (9.0–6.0 kyr cal BP). Secondly, the SB2 core was obtained from the western shore of Lake Banyoles, 160 m away from the present lakeshore. This core is 370 cm long, with a continuous analysed pollen record between 174 and 281 cm depth, providing data for the period 9.0–3.3 kyr cal BP [38,39].

3.1.4. La Draga (NE Iberia)

The site of La Draga, located on the eastern shore of Lake Banyoles (173 m asl) (Girona, NE Spain) (Figure 1), was one of the earliest farming communities in open-air settlements in

NE Iberia, dated to 7.27–6.7 kyr cal BP [40,41]. A total of 825 m$^2$ have been excavated since the start of archaeological fieldwork at La Draga in 1991. In 1991–2005, the archaeological excavations focused on Sectors A, B, and C (Figure 2). Sector A is an emerged sector; Sector B is partially waterlogged; and Sector C is totally underwater [42,43]. In recent work, Sector D (with the same conditions as Sector B) was excavated in 2010–2013, and a new area in Sector A has been excavated since 2013 (Figure 2). Two different construction phases during the early Neolithic occupation were documented in Sector B-D [44]: Phase I (7.27–6.93 kyr cal BP) is characterized by the collapse of wooden structures, which have been preserved in an anoxic environment; Phase II (7.16–6.75 kyr cal BP) is represented by several pavements of travertine stone where domestic activities were carried out and where organic matter is only preserved in a charred state. In Sector A, these two phases evidenced in Sector B-D are not clearly visible and, in their place, a palimpsest of travertine stone structures, archaeological remains and organic clayish sediments appears. The structures (E254, E258, E261, E263) documented at Sector A are dated in 7.1–6.7 kyr cal BP, coetaneous with Phase II in sector B-D [41]. These structures are partially covered by terrigenous mineral sediments (UE2002), which would have been deposited after/during the abandonment of this sector.

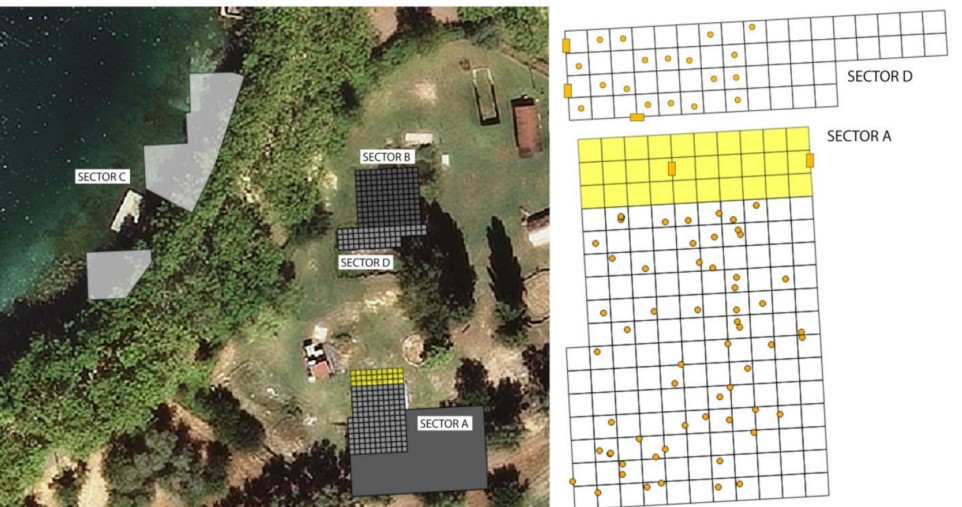

**Figure 2.** Plan view of sectors at La Draga. In dark grey, sectors excavated between 1991 and 2005 (**A**,**B**).; in light grey, in 1994-2005 (**C**), 2010–2013 (**D**) and 2013–2017 (**A**); in yellow ongoing excavation (**A**). Location of analysed pollen samples in Sectors A and D: the circles represent pollen samples in spatial analysis, the rectangles represent the archaeological profiles analysed.

The sampling strategy for pollen analysis at La Draga was designed to achieve two different objectives: the first, to obtain a diachronic view of landscape evolution in the different occupation phases; the second, to develop a spatial analysis of pollen and non-pollen palynomorphs in order to identify different environments and uses of space within the settlement. Thus, the analysis of archaeological profiles in Sector A [45,46], C [47] and D [15] enabled the reconstruction of landscape evolution in the different occupation phases, as well as providing information about site formation processes. On the other hand, the horizontal sampling consisted of retrieving sediment samples from different structures and archaeological layers during the excavation in both Sectors D [48] and A [49].

### 3.2. Palynology, the Study of Pollen and Non-Pollen Palynomorphs (NPP)

Palynology is defined as the study of pollen and spores within sediments in order to reconstruct vegetation history and its interaction with environmental and climatic changes, as well as with human activities, assuming a relationship between the amount of pollen grains of taxa deposited in sediments and the individuals of these taxa in surrounding vegetation [50–52]. Pollen analysis at archaeological sites or in nearby natural deposits is an essential source of information for archaeoecological research, given that many human

activities alter the pollen record by inputs of plants to the settlement or by causing disturbance processes in the surrounding forests. In that sense, palynology gains importance at a local scale, providing information about deforestation processes, the use or consumption of certain plants (pollen grains) or fungi (fungal spores, NPP), the introduction of exotic species or the adoption of cultivars (mainly cereals and legumes in Neolithic chronologies), the presence of flocks (spores of coprophilous fungi) and detailed hydrological (dryness/wetness, algal remains, NPP) and geomorphological (soil erosion indicators, NPP) information [53,54].

Lakes, lagoons and bogs, where taphonomic processes are less frequent [55], are the most appropriate deposits with which to approach the original pollen rain. In contrast, various taphonomic processes occur in aerial/subaerial mineral deposits such as archaeological sites where they can deform the original deposition of pollen and non-pollen palynomorphs (NPP). In other words, while pollen deposition in natural deposits is presumably homogenous, significant differences can be attested in pollen abundance within small areas, and this is more likely in records affected by taphonomic agents, especially in archaeological contexts. Therefore, several taphonomic processes that distort the original pollen rain [56] are even more accentuated at archaeological sites, which cannot be considered the most ideal archives for palaeoenvironmental reconstruction. However, the study of pollen at archaeological sites provides answers to historical research questions [57] and contributes essential data to reconstruct site formation processes and social use of space within settlements [48]. In that sense, the application of palynological analyses in archaeological deposits (archaeopalynology) can obtain evidence about socioeconomic practices (crops, gathered plants, storage, stabling of flocks) and human impact in the surroundings (deforestation).

In this study, pollen samples were obtained exclusively from organic clay-richunits (in both natural and archaeological deposits) due to low pollen concentrations and taphonomic biases in sandy layers and in oxidized silts. Samples were processed following standard methods [58,59]: treatment with HCl and NaOH, flotation in Thoulet heavy liquid, treatment in HF, and finally mounting in glycerine. 300–400 pollen grains of terrestrial taxa (the pollen sum was lower in someintervals due to low pollen concentrations) were counted using an Olympus Bx43 microscope fitted with x10 oculars and x40/60objectives. Hygrophytic and aquatic plants (Cyperaceae, *Typha latifolia* and *Typha/Sparganium*, *Myriophyllum*, *Nuphar*, *Nymphaea*) and terrestrial local plants in Corsican coastal lagoons (Amaranthaceae-Chenopodioideae) were excluded from the pollen sum to avoid over-representation bylocal taxa. All pollen types are defined according to [60] and Cerealia-type was defined according to the morphometriccriteria of [55] (pollen grain >40 mm, pore diameter >8 mm). The identification of Non-pollen palynomorphs (NPPs) followed [15,61–64]. Percentage pollen diagrams were created using Tilia software [65] and pollen zones were defined using stratigraphically constrained cluster analysis (CONISS) [66].

Following the ORDP, and aiming to improve and maximize the access to the research data, pollen datasets from natural deposits presented in this work have been submitted to the European Pollen Database (EPD, http://www.europeanpollendatabase.net/) (accessed on 27 September 2017 (SB2 Banyoles), 4 February 2020 (Saint Florent) and 15 April 2020 (Piantarella)) [33,34,39]

### 3.3. Numerical Analysis

The integration of spatial analyses allows for a better comprehension of formation processes within the site and a better understanding of the use of space within the settlement [67,68]; crucial information given the difficulties of reconstruction of domestic structures and spaces in open-air pile dwelling sites. The purpose of the analysis arises out of the need to interrelate the different variables to characterise the uses given to the space. With this aim, a two-step procedure is involved: running a Correspondence Analysis using the algorithm "Detrended" (DCA) in order to avoid distortion in ordination and overemphasis of rare taxa; and the creation of interpolations of the row scores obtained

in the DCA for Axis 1 and 2 on the x, y coordinates, so as to localise the particular cells with statistically significant discriminant values [69]. For these interpolations the algorithm used was Inverse Distance Weighting (IDW) and the software was Past v. 3.12. Mapping techniques were also applied in order to visualize the most probable distribution of the different taxa analysed in the space. This paper provides the interpolation for Cerealia-t by the application of the Kriging algorithm. All spatial interpolations performed were automatically adjusted to the theoretical models of a Gaussian or an Exponential, considering the optimal models to fit the variogram to the data analysed [48]. The software used was RockWorks17.

## 4. Results

### 4.1. Piantarella (South Corsica, France)

*Erica* was dominant in the landscape from ~7.4 kyr cal BP to the present (maximum of 64% at ~7.4 kyr cal BP, Zone Piant-1/A, Figure 3), with phases of regression during 7.3–7.2 kyr cal BP in Piantarella 1 (Zone Piant-1/B), and during 6.6–6.4 kyr cal BPin Piantarella 3 (Zone Piant-3/A, Figure 3). Low values of AP (~10–25%), predominated by *Pinus*, with very low values in sclerophyllous, riparian and broadleaf deciduous trees in Piantarella 1. Cerealia-t is continuously present in Piantarella 1 (7.35–7.2 kyr cal BP), coinciding with regression in *Erica*, expansion of herbs, occurrence of spores of coprophilous fungi (mainly *Sordaria* and *Sporormiella* [24]), and in some cases, peaks in Asteraceae and fungal indicators of soil erosion. In Piantarella 3, the occurrence of Cerealia-t and coprophilous fungi coincides with falls in *Erica* values in 6.6, 6.45, and 5.6 kyr cal BP and with regression of sclerophyllous trees in 5.35–5.3 kyr cal BP. Sclerophyllous trees show their highest values in Pollen Zones Piant-3/B (5.9–5.3 kyr cal BP) and Piant-2/A (4.7–4.6 kyr cal BP); and in Piant-2/B (0.4 kyr cal BP to present) *Erica* maquis is still predominant, but now in a more open landscape, by the expansion of herbs (Figure 3), and, most likely, with a significant role of fire given the peak in lignicolous/carbonicolous fungi. Amaranthaceae-Chenopodioideae show their highest values in zones Piant-1/A, Piant-2/A, and Piant-2/B, and, on the other hand, the highest values in *Isoetes* occur in zones Piant-1/B, Piant-3/A, and Piant-3/B (Figure 3). Freshwater algae show peaks in Piant-3/A and Piant-2/B and aquatic and hygrophilous plants show a continuous curve with moderate values in Piant-1 and Piant-3, and almost disappear in Piant-2, from 4.7 kyr cal BP onwards.

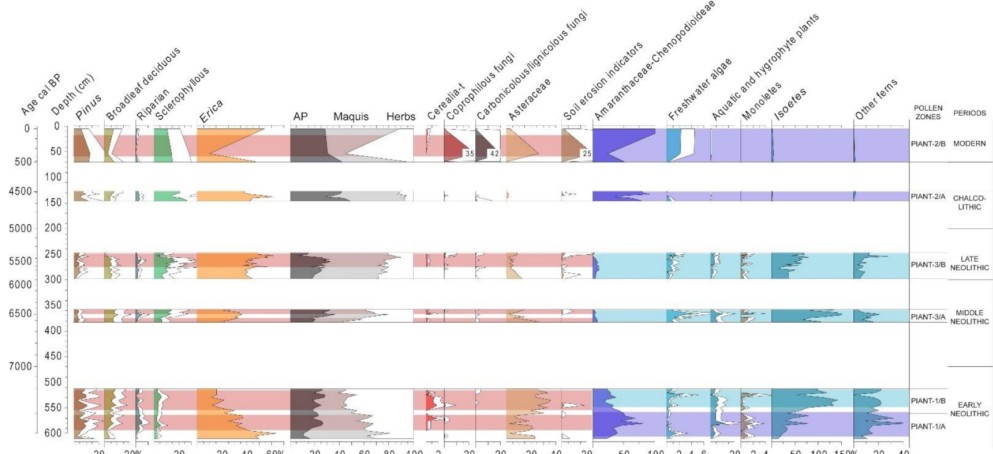

**Figure 3.** Percentage pollen and NPP diagram of selected taxa and categories for the Piantarella record (South Corsica). Red shading indicates phases of human impact, dark blue increases in salinity, and sky-blue freshwater phases. Categories (also valid for Figure 4): broadleaf deciduous (deciduous *Quercus, Corylus, Fagus, Tilia, Carpinus, Acer*); riparian (*Alnus* cf. *glutinosa, Alnus* cf. *suaveolens, Ulmus, Fraxinus, Salix*); sclerophyllous (*Quercus ilex-coccifera, Quercus suber*-type, *Olea, Phillyrea,* cf. *Juniperus*); maquis (*Erica, Arbutus unedo, Pistacia,* Cistaceae, *Ephedra, Thymelaea, Asphodelus*); Coprophilous fungi (*Sordaria*-type, *Cercophora*-type, *Podospora*-type, *Sporormiella, Delitschia*), Carbonicolous-lignicolous fungi

(*Gelasinospora* HdV-2, *Neurospora*, *Coniochaeta* cf. *ligniaria*, *C. xilariispora* HdV-6, *Kretzschmaria deusta*, types UAB-9, UAB-11, UAB-12 and UAB-38); Asteraceae (A. liguliflorae, A. tubuliflorae, *Aster*-type, *Cirsium*-type, *Centaurea*); soil-erosion indicators (*Glomus*, UAB-8); aquatic and hygrophyte plants (Cyperaceae, *Typha*, *Typha-Sparganium*, *Myriophyllum*, *Nuphar*); freshwater algae (*Spirogyra*, *Spirogyra* HdV-210, *Spirogyra reticulata*, *Zygnema*-type, *Pediastrum*, *Mougeotia*, *Debarya*, *Closterium*), Other ferns (*Polypodium*, *Pteridium*, *Osmunda*-type, *Ophioglossum*-type and other trilete spores).

### 4.2. Saint Florent (North Corsica, France)

The riparian forest (mainly *Alnus* spp. [24]) predominates in zones SF-A1 and SF-A2, showing also high values in *Pinus*, sclerophyllous forests and *Erica*, and low values of broadleaf deciduous vegetation (Figure 4). Sclerophyllous forests expanded in the Late Holocene, from 4.2 kyr cal BP onwards, while riparian forests experienced a regression. The expansion of herbs, and the occurrence of Cerealia-t, spores of coprophilous fungi and soil-erosion indicators coincide in ~5.9–5.8 and ~5.5 kyr cal BP (Zone SF-A1), in ~5.2–5.0 and ~4.9–4.5 kyr cal BP (Zone SF-A2), ~4.0–3.5 and ~2.0 kyr cal BP (Zone SF-B1) and in 1.0 kyr to present (Zone SF-B2) (Figure 4). The AP curve shows high values during the whole sequence until Zone SF-B2, when AP falls abruptly. Very low values of Amaranthaceae-Chenopodioideae in SF-A1/A2 is followed by high values in SF-B1/B2, in contrasting dynamics to *Isoetes* and other ferns, which show their highest values in ~6.3–4.5 kyr cal BP. Finally, aquatic and hygrophilous plants and freshwater algae present their highest values in SF-B2 (Figure 4).

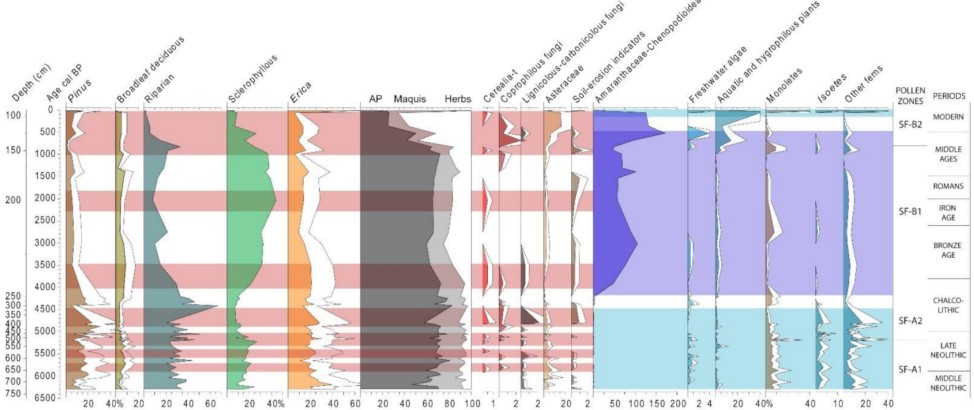

**Figure 4.** Percentage pollen and NPP diagram of selected taxa and categories in the Saint Florent record (North Corsica). Redshading indicates phases of human impact, dark blue increases in salinity, and sky-blue freshwater phases. For categories, see Figure 3.

### 4.3. Lake Banyoles (Girona, NE Iberia)

Lake Banyoles (LB) sequence shows the expansion of forests from the beginning of the Holocene with a decreasing trend of the grasslands that formed steppes in the Late Glacial period [37]. In 9.0 kyr cal BP, the expansion of forests was completed, reaching 90% of AP with the dominance of broadleaf deciduous trees, mainly deciduous *Quercus* and *Corylus* (discontinuous silhouettes in Figure 5). In that sense, the deciduous forest optimum occurred at about 9.0–7.6 kyr cal BP, although it may be extended until the end of the sequence (6.1 kyr cal BP), after some centuries of regression of oak forests. The start of the regression of deciduous *Quercus* occurred in ~8.2 kyr cal BP and was consolidated after 7.6–7.4 kyr cal BP. This oak forest regression coincides with the expansion of *Abies*, which appeared in the area in ~8.5 kyr cal BP and displays its maximum values in the phase 8.0–6.6 kyr cal BP. When the first farming societies reached the Lake Banyoles area (7.4–7.0 kyr cal BP) a slight expansion of non-arboreal pollen (NAP) is attested in the context of episodes of regression of the oak forest.

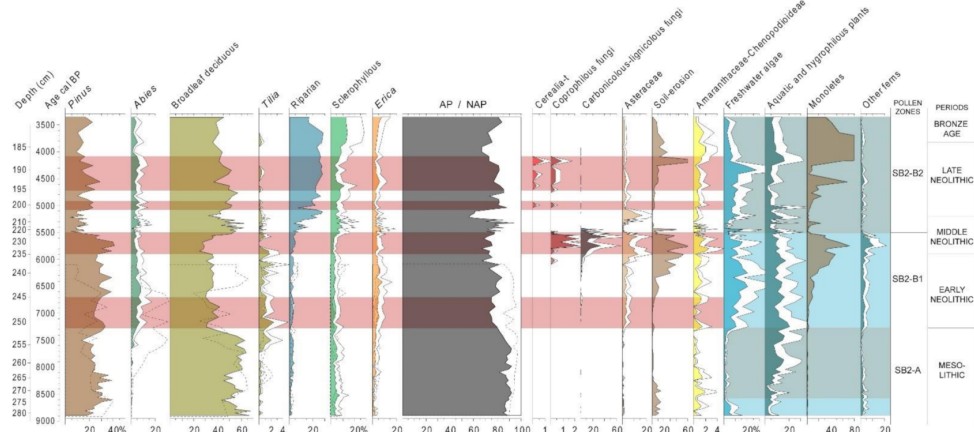

**Figure 5.** Percentage pollen and NPP diagram of selected taxa and categories for SB2 and LB (discontinuous silhouettes) records (Banyoles, Girona, NE Iberia. Shadings in red show human impact, dark greenish blue marks peaty phases, sky blue, aquatic phases. Categories: broadleaf deciduous (*Quercus* deciduous, *Corylus*, *Fagus*); Riparian (*Alnus*, *Salix*, *Fraxinus*, *Ulmus*); sclerophyllous (*Quercus ilex-coccifera*, *Olea*, *Phillyrea*); coprophilous fungi (*Sordaria*-type, *Podospora*-type, *Cercophora*-type, *Rhytidospora*); carbonicolous-lignicolous fungi (*Gelasinospora* HdV-2, *Coniochaeta* cf. *ligniaria*, *Kretzschmaria deusta*, types UAB-9, UAB-10, UAB-11, UAB-12, UAB-17); Asteraceae (Ast. liguliflorae, Ast. tubuliflorae, *Cirsium*-t, *Aster*-t, *Centaurea*); Soil erosion indicators (*Glomus*, HdV-361; UAB-8; UAB-30); Freshwater algae (*Spirogyra*, *Zygnema*, *Closterium*, *Mougeotia*); Aquatic and hygrophyte plants (Cyperaceae, *Typha latifolia*, *Typha-Sparganium*, *Nymphaea*); other ferns (*Pteridium*, *Polypodium*).

From its start, the SB2 sequence shows the dominance of broadleaf deciduous forests coinciding with the optimum evidenced in the LB sequence from 9.0 kyr cal BP onwards. Despite episodes and phases of oak forest regression in 7.25–5.55 kyr cal BP (Zone SB2-B1, Figure 5), coinciding with the establishment of the first Neolithic communities in the area, the optimum of broadleaf deciduous forests is maintained until the end of the sequence, indicating a persistence of oak forests until the onset of the Late Holocene. The oak decline coincides with an expansion of secondary trees such as *Pinus* or *Tilia* and NAP (Figure 5). Noteworthy as well is the first arrival of *Abies* in the area in 8.5–8.0 kyr cal BP (Zone SB2-A), appearing with a continuous curve after 7.6 kyr cal BP, thus coinciding with the LB sequence. In ~5.9–5.5 kyr cal BP, high values in spores of coprophilous and lignicolous-carbonicolous fungi, Asteraceae and soil-erosion indicators are documented. Then, the earliest evidence of Cerealia-t is attested in ~4.98, 4.63, 4.41 and 4.17 kyr cal BP, coinciding with spores of coprophilous fungi (Figure 5). In Zone A (~8.9–7.3 kyr cal BP), increasing values in aquatic and hygrophilous plants are related to decreasing values in freshwater algae and low values of riparian trees. In Zone B1 (~7.3–5.5 kyr cal BP), there is a sharp increase in freshwater algae and peaks in monolete spores and other ferns. Then, Zone B2 (~5.5–3.4 kyr cal BP) shows an expansion in riparian trees (mainly *Alnus* spp. [38]), a decline in aquatic and hygrophilous plants and increasing values with high peaks in monoletes (Figure 5), indicating the establishment of an alder carr environment [64].

### 4.4. La Draga (Girona, NE Iberia)

#### 4.4.1. Archaeological Profiles

Pollen analysis in archaeological profiles at La Draga provided robust data to characterize environmental evolution during the different occupation phases and the moments prior and posterior to the Neolithic settlement [15]. This analysis confirms that on their arrival, the first farmers found dense forests predominated by broadleaf deciduous trees (dec. *Quercus*, *Corylus*), with riparian forests next to the settlement (*Fraxinus*, *Salix*, *Ulmus*, *Alnus*), a secondary role of conifers (*Pinus* and *Abies*) and a slight signal of regional sclerophyllous forests (mainly *Quercus ilex-coccifera*) and Mediterranean maquis (*Erica*, Figure 6).

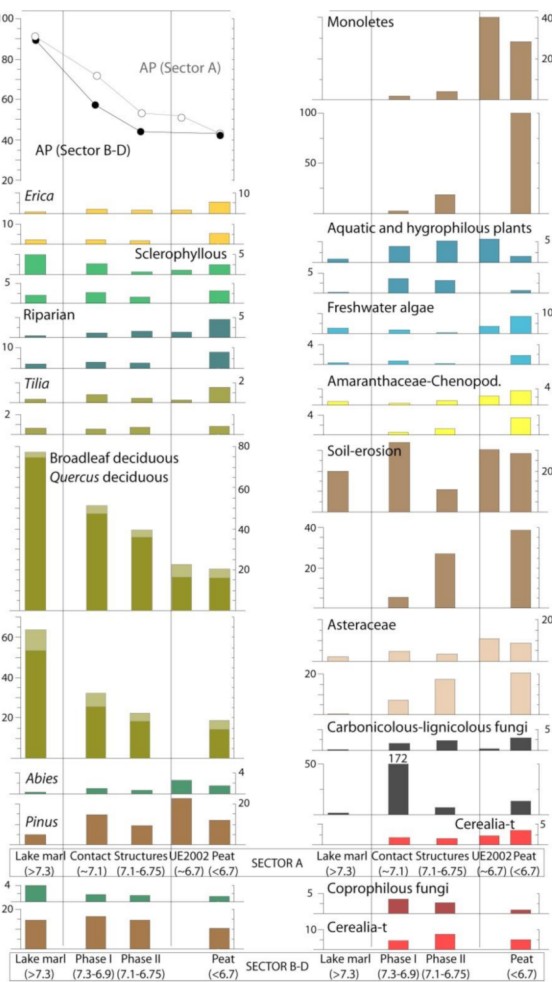

**Figure 6.** Percentage pollen and NPP diagram for La Draga archaeological profiles. Categories: broadleaf deciduous (*Quercus* deciduous, *Corylus*); Riparian (*Alnus*, *Fraxinus*, *Salix*, *Ulmus*); sclerophyllous (*Quercus ilex-coccifera*, *Olea*, *Phillyrea*); coprophilous fungi (*Sordaria*-type, *Podospora*-type, *Cercophora*-type, *Rhytidospora*, *Apiosordaria verruculosa*, UAB-2, UAB-34B, UAB-36); carbonicolous-lignicolous fungi (*Chaetomium*, *Gelasinospora* HdV-1/2, *Coniochaeta* cf. *ligniaria*, *Coniochaeta xilariispora* HdV-6, *Kretzschmaria deusta*, *Neurospora*, UAB types 4, 9, 10, 11, 12, 17, 38); Asteraceae (Ast. liguliflorae, Ast. tubuliflorae, *Cirsium*-t, *Aster*-t, *Centaurea*); Soil erosion indicators (*Glomus*, HdV-182, HdV-361, UAB types 8, 22, 30, 46, 47, 48, 50); Freshwater algae (*Spirogyra*, *Spirogyra* HdV-210, *Spirogyra reticulate*, *Zygnema*, *Mougeotia*); Aquatic and hygrophyte plants (Cyperaceae, *Typha latifolia*, *Typha-Sparganium*, *Nymphaea*); other ferns (*Pteridium*, *Polypodium*).

From Phase I in Sector B-D (~7.3–6.9 kyr cal BP), clear signs of deforestation led by Neolithic communities are evident in the abrupt fall of *Quercus* deciduous values. This is also attested in the first centimetres of lake marl (contact with archaeological sediments, Figure 6) in Sector A. The occurrence of Cerealia-t, spores of coprophilous fungi and high values of herbs show the local impact of human activities in the pollen record of different archaeological layers and structures. A large accumulation of spores of carbonicolous-lignicolous fungi was recorded in Phase I in Sector B-D, and general scarce values in Sector A. While the highest values of aquatic and hygrophilous plants occur during the occupation at La Draga, they decline after the abandonment in favour of riparian trees. High values in Asteraceae occurs in Phase II in Sector B-D and in layers UE2002 and post-abandonment peat, while soil erosion indicators are present in high values in the lake marl in Sector A, and in Phase II and peat in Sector B-D (Figure 6). Monolete spores and freshwater algae show their highest values in peat layers after the abandonment of the site.

4.4.2. Spatial Distribution of Pollen and NPP

In general, most of the pollen and NPP taxa displayed heterogeneity in their frequencies in different samples (i.e., *Pinus*: 3–39.4%; Cerealia-t: 0.3–10%), proving the suitability of spatial analysis in La Draga pollen record. In Sector B–D (Figure 7), the DCA shows an Eigenvalue of 0.1684 (81.51%) for Axis 1 and 0.0274 (13.26%) for Axis 2. The IDW represents the spatial distribution of Axis 1 and 2, showing an east-west gradient for Axis 1. Thus, in the east higher richness (diversity) was recorded and many taxa present maximum values, while in the west few taxa dominate the record with high values. In the case of Axis 2, a concentration of some taxa was identified in the north-eastern area. The DCA results for pollen and NPP in Sector A (Figure 8) show that Axis 1 and 2 explain 84.84% of the variance, with an Eigenvalue of 0.2737 (71.2%) in Axis 1 and 0.05218 (13.6%) in Axis 2. Positive values in Axis 1 are defined by predominant taxa in mineral sediments (determined by loss on ignition, [49]) deposited by soil erosion episodes (*Pinus*, *Abies*, *Pteridium*, *Ophioglossum*-t). Positive values in Axis 2 are associated with the positive correlation of anthropogenic taxa (Fabaceae, *Verbena officinalis*, *Vitis*, *Aster*-t, *Hypecoum*-t, Cerealia-t). The IDW represents the spatial distribution of Axis 1 and 2 (Figure 8), showing the highest values in Axis 1 associated with UE2002. In the case of Axis 2, a concentration of anthropogenic taxa is identified in the north-eastern area, associated with the structure E263. Finally, Kriging interpolation provided a two-dimensional representation of Cerealia-t pollen distribution in space and an estimation of predicted values between measured points (Figure 9). Exponential and Gaussian models were the best fitting models, with 0.88–0.99 values in r$^2$ [48,49] and the spatial accumulation of Cerealia-t being identified in the south-western corner of Sector B-D and in structures E261 and E263 in Sector A (Figure 9).

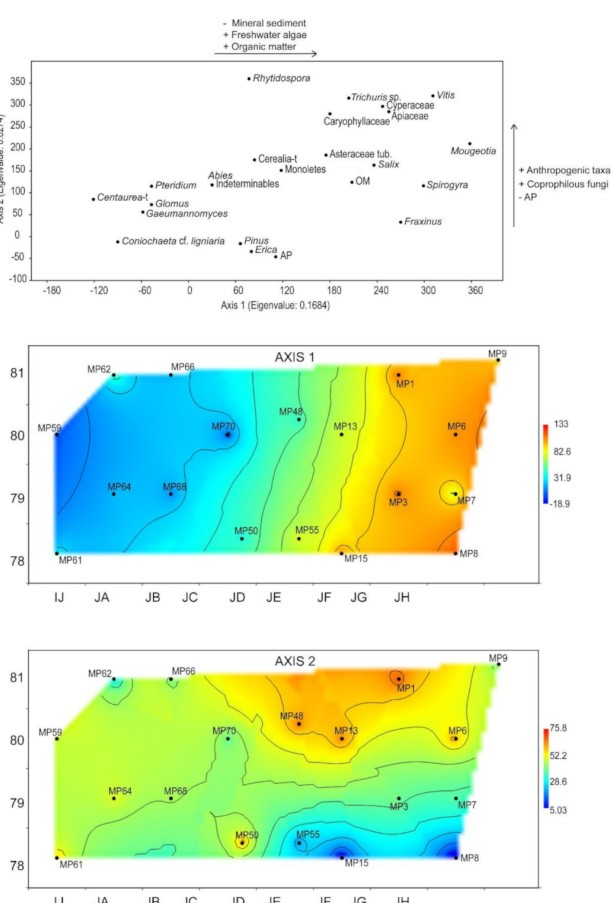

**Figure 7.** Graph showing the results of Detrended Correspondence Analysis and interpolation by Inverse distance weighting analysis (IDWA) (past software) in Sector D, plotting the row scores of the first and second axis on the x, y coordinates.

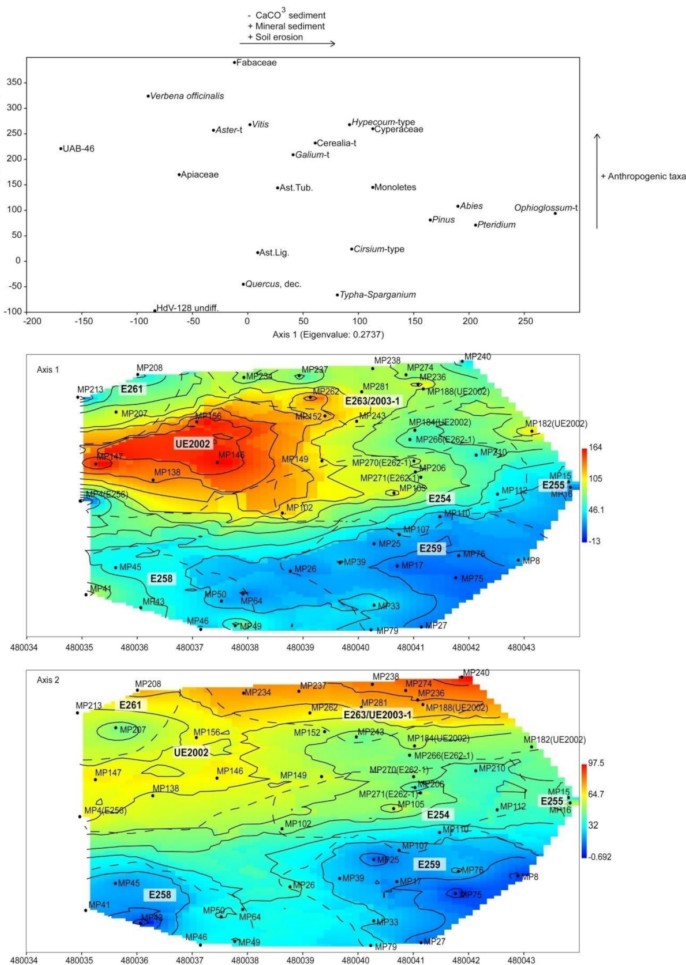

**Figure 8.** Graph showing the results of Detrended Correspondence Analysis, and interpolation by Inverse distance weighting analysis (IDWA) (Past software) in Sector A, plotting the row scores of the first and second axis on the x, y coordinates.

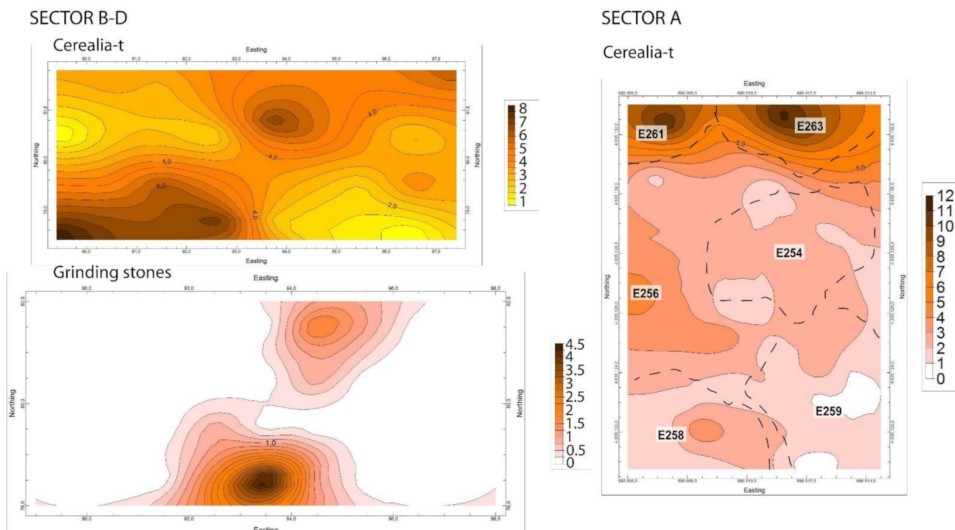

**Figure 9.** Kriging interpolation maps for Cerealia-t (Sectors A—right and D—left) and grinding stones (Sector D).

## 5. Discussion

*5.1. Mid-Holocene Vegetation History and Neolithic Landscape Transformation in the Western Mediterranean*

The palynological study of Piantarella and Saint Florent sequences provided new insights into the Corsican landscapes prior to their transformation by humans. As shown in Piantarella (South Corsica), the landscape during the Early Neolithic was predominated by *Erica*, which could have formed forests during the Early Holocene, but the continuous impact of human activities would have converted it to Mediterranean scrubland [24,70]. In that sense, the phases of regression in *Erica* in 7.3–7.2 kyr cal BP in Piantarella (Figure 3) are clear evidence of the onset of this process of landscape transformation in which Early Neolithic communities played a major role. This regression of Mediterranean scrublands led to the expansion of herbs, peaks in Asteraceae, occurrence of Cerealia-t and spores of coprophilous fungi, as well as some peaks in soil-erosion indicators (Figure 3). Thus, the Piantarella 1 record is notable for being the earliest evidence of the impact of farming activities in southern Corsica [24], and could be related with nearby Early Neolithic sites, such as Araguina Sennola (7.8–7.0 kyr cal BP, [71], Longone (7.5–6.9 kyr cal BP, [19]) and/or Abri du Goulet (EN without [14]C dating).

In Saint Florent (North Corsica), the pollen record indicates denser vegetation, and the relatively high values of *Erica* (lower than in Piantarella) would be related to Mediterranean scrubland growing on drier soils or at lower elevations in the context of the predominance of evergreen oak forest at a regional scale and riparian forests at local scale (Figure 4). These differences in the vegetation would be related to a more humid climate in North Corsica, but the fact that the Saint Florent sequence starts at 6.3 kyr cal BP does not allow an assessment of the existing landscape when the first farming communities became established in the region, and the low values of *Erica* could be the effect of previous human impacts in North Corsica. For that reason, future research will focus on drilling new cores in the area to reach older chronologies and reconstruct vegetation history from the Early Neolithic.

In any case, the landscape in North Corsica was dominated by forests until the Late Holocene, with relativelyhigh AP values (60–80%),and the existence of open vegetation being the result of human impacts in historical times (0.7−−0.02 kyr cal BP, Figure 4). However, phases of human impact are characterised during the Late Neolithic-Chalcolithic by the occurrence of Cerealia-type pollen and spores of coprophilous fungi, in the context of forest decline (riparian forests in ~5.9–5.8 kyr cal BP, and mesic and sclerophyllous forests in ~5.2–5.0 kyr cal BP, Zone SF-A2, Figure 4) and the expansion of maquis. A clear difference is seen between North and South Corsica; while Piantarella shows how human impact affected *Erica*-dominated maquis, landscape transformation activities in Saint Florent affected mainly riparian and sclerophyllous forests, and the expansion of maquis acts as a signal of human impact. Both regions indeed coincide in the clear evidence of the impact of farming activities in coastal areas, suggesting that coastal wetlands in Corsica were highly valued by the first farming communities for their settlements and agricultural and husbandry activities from the Early Neolithic and in different phases in the Middle and Late Neolithic, Chalcolithic, Bronze Age and, thereafter in historical periods.

The results from palynological studies in Lake Banyoles show that the first farmers found a region covered by dense broadleaf deciduous forests (Figure 5), and both LB and SB2 records show an abrupt decline in oak forest in ~7.4–7.3 kyr cal BP, coinciding with the early Neolithic settlement of La Draga. Although the evidence of deforestation during the early stages of Neolithisation in Iberia is often associated with farming activities, in Lake Banyoles the absence of Cerealia-t pollen and spores of coprophilous fungi is noticeable, but not exclusive in the Iberian context [72–78]. In that sense, the expansion of the first farming societies and the practice of intensive agriculture [79] implied limited impact on the landscape. However, sedentary life in permanent settlements and the practice of more intensive productive activities increasingly reiterated in the territory caused deforestation processes or, at least, small-scale forest modifications from the Neolithic onwards [23].

In that sense, Neolithic landscape transformation was expressed differently in the regions studied here. While, human activities caused the regression of maquis in South Corsica, leading to an open landscape, in North Corsica they affected riparian and sclerophyllous forests, and in NE Iberia the dominant broadleaf deciduous forests. In coastal wetlands in Corsica, evidence of Cerealia-t is clear from the Early Neolithic and repeatedly along the Neolithic. In NE Iberia, evidence of agriculture was invisible in Lake Banyoles during the Early Neolithic, restricted to the immediate surroundings of the archaeological site of La Draga [15,48]. This could suggest either a very local impact of agriculture in Corsican coastal wetlands, or the fact that these lagoons or wetland areas provide a pollen signal from a smaller catchment area when compared with large lakes [80], as is the case of Lake Banyoles, whose pollen record reflects the regional landscape and so that it becomes more difficult to detect the signal of agriculture. In that sense, Cerealia-t is attested in Lake Banyoles after the transition from aquatic (larger catchment area) to an alder carr peaty environment (smaller catchment area) from 5.5 kyr cal BP onwards [38,64].

Apart from evidence of agricultural activities, spores of coprophilous fungi are a relevant proxy to reconstruct the impact of husbandry, a factor to be considered in vegetation shifts from the Neolithic onwards [77,81,82], and well attested in this study. Another element which differs is fire, almost absent in NE Iberia [38] and, until further palaeofire studies are conducted in coastal wetlands, it seems evident that fire (whether natural or anthropogenic) played a major role in landscape configuration in Corsica due to the continuous presence of macro and micro charcoal in pollen samples [24]. In the Lake Banyoles area, despite the absence of Cerealia-t and spores of coprophilous fungi in pollen records during the Early Neolithic, the bioarchaeological record at La Draga enables an interpretation of the regression of broadleaf deciduous forests (mainly *Quercus* deciduous) as the result of deforestation to acquire raw materials for construction (oak wood) and firewood [23,83]. Finally, another useful proxy to detect human impact is the occurrence of soil erosion episodes (reflected in the occurrence of indicative fungal spores or even high peaks in Asteraceae), often induced by climate change or heavy rainfall, but amplified in phases of disturbed landscapes by deforestation, as attested during the Neolithic in SB2 record, Piantarella, Saint Florent, and at La Draga in Phase II, consistent with similar processes evidenced in the Mediterranean region, as on the Eastern coast of Iberia, where the combination of fire activity, the impact of farming and occurrence of soil erosion episodes have been documented during the Neolithic [84].

### 5.2. Local Palaeoenvironmental Evolution in Lakes and Wetlands, the Value of Non-Pollen Palynomorphs (NPP)

The archaeoecological study of human-environment interactions involves different scales transcending in-site archaeology, from settlement surroundings to regional landscapes. As shown in the previous section, palynology is able to reconstruct vegetation history and attests signs of human impact in the landscape on regional or extra-local scales. However, the study of short-dispersal pollen taxa and non-pollen palynomorphs allow the strictly local palaeoenvironmental evolution in lakes, lagoons or wetland areas to be assessed.

Regarding pollen, the occurrence or overrepresentation of animal-borne or water-borne taxa suggest the local presence of these plants. This is the case of aquatic taxa, such as *Nymphaea*, *Nuphar* or *Myriophyllum*, suggesting the existence of shallow waters in the analysed records; and terrestrial taxa, such as Asteraceae, showing peaks in Piantarella (Figure 3), Saint Florent (Figure 4) and Lake Banyoles (Figure 5). High values of Asteraceae may be interpreted as their input in the context of soil erosion episodes, bringing to the lakes or wetlands eroded terrestrial materials (mineral soils, charcoals, plants) from upland; or alternatively as evidence of edaphisation, i.e., the transition from humid substrates to dry terrestrial ones. Edaphisation is commonly accompanied by low pollen concentration, low taxonomic richness and overrepresentation of Asteraceae. However, peaks of this taxon occur in these sites in rich samples without signs of taphonomic bias, so Asteraceae acts as an indicator of soil erosion and of the expansion of open areas in the site catchment

area. Although Amaranthaceae-Chenopodioideae is an anemophilous (wind-dispersed) pollen taxon, its overrepresentation in coastal lagoons is a clear sign of local occurrence in the development of saline marshlands. In that sense, Amaranthaceae-Chenopodioideae indicates marine incursions in Piantarella 1 (Zone Piant-1/A, ~7.4 kyr cal BP) and the establishment of saline marshlands during the Late Holocene in South Corsica (Piantarella, from 4.6 kyr cal BP onwards, Figure 3) and North Corsica (Saint Florent, from 4.1 kyr cal BP onwards, Figure 4), consistent with other Mediterranean coastal sites [84–91], in the context of drier climate and increasing salinity in coastal wetlands. However, the occurrence of scarce values of Amaranthaceae-Chenopodioideae in an inland freshwater lake, as is the case of Lake Banyoles (Figure 5), has to be interpreted in different terms, in which it is considered a ruderal plant often showing peaks in periods of higher human impact.

Apart from the abovementioned pollen taxa, the analysis of non-pollen palynomorphs provides robust data on the palaeoenvironmental evolution at local scale from a varied assemblage of microremains: spores of ferns, algae, fungi, and other unclassified palynomorphs. Among ferns, *Isoetes* was the best represented in Piantarella in 7.4–5.3 kyr cal BP (Figure 3) and shows a continuous curve in the first part of Saint Florent (6.3–4.2 kyr cal BP, Figure 4), indicating the existence of shallow freshwater. On the other hand, high values of monolete fern spores are related to either soil erosion in aquatic environments or their local growth in sub-aerial substrates [64]. In that sense, the occurrence of monoletes in aquatic phases, such as in Zones Piant-1/A and B, Piant-3/A and B, SF-A1 and A2 and SB2-B1, has to be understood in the context of soil erosion episodes associated with the expansion of open landscapes by human-induced deforestation; in the case of sub-aerial environments, such as marshlands (SF-B1) and alder carr (SB2-B2), monoletes indicate drier substrates in relation to climate changes in the transition to the Late Holocene. In addition, the identification of algal microfossils in pollen slides provided significant data to comprehend the evolution of water bodies in lakes and wetlands. The category 'freshwater algae' (Figures 3–5) includes different taxa (i.e., *Spirogyra*, *Mougeotia*, Zygnemataceae) and, evidently, the detailed study of their distinct dynamics would provide more detailed information on water limnology [64]. However, the sum of the different algal microfossils served as an indicator of alternation between phases of marine/freshwater in Corsican records (Figure 3; Figure 4) and between sub-aerial peaty and clayish aquatic phases in Lake Banyoles (Figure 5). Finally, the occurrence of different types of fungal spores enabled the identification of frequentation of animals (coprophilous fungi mainly related to large herbivores) in the surroundings of water bodies, phases of more intense fire activity and/or forest perturbation (carbonicolous-lignicolous fungi), and episodes of soil erosion (chlamydospores of *Glomus* [92]; and types linked with mineral soils, i.e., UAB-8 [15]).

### 5.3. Archaeopalynology: How Pollen and NPP Can Provide Relevant Data for the Study of Plant Uses, Social Use of Space and Site Formation Processes

5.3.1. Site Formation Processes and Environmental Evolution

The analysis of pollen and non-pollen palynomorphs in archaeological profiles gives a picture of environmental evolution within the site and in the nearby surroundings. In addition, the pollen content of different archaeological layers and structures provides significant information to reconstruct site formation processes and, in combination with geomorphological studies, enables the identification of depositional and post-depositional events in archaeological contexts.

In La Draga, the lake marl offers an insight into the landscape prior to the Neolithic occupation, showing that there was a dense oak forest (*Quercus* deciduous), presence of pines and fir in the surrounding mountains (*Pinus* and *Abies*), and riparian forests at local scale (Figure 6). Once Neolithic communities settled at La Draga (Phase I, Figure 6) evident vegetation shifts occur: a fall in *Quercus* deciduous, the overrepresentation of herbs, the expansion of hygrophilous plants indicating open spaces at local scale, and a slight expansion of pines. The contact layer (first 2 cm) between lake marl and archaeological sediments in Sector A is noteworthy, as the evidence of human impact is already detected, suggesting that in this emerged sector, the lake marl was exposed (confirmed by high values

of soil erosion indicating sub-aerial or terrestrial environments) and used as the occupation surface where humans carried out their activities (clearly evidenced, for example, in the presence of Cerealia-t pollen, Figure 6). In Phase II (Sector B-D) and in Sector A structures, the signal of landscape transformation is accentuated, showing both lower values of *Quercus* deciduous and high values of Cerealia-t. In Sector B-D, the occurrence of high values of coprophilous spores in both phases contrast with their absence in the analysed profiles in Sector A and their rare occurrence in the spatial analysis [49,93]. Once the settlement was abandoned, a peaty layer formed and riparian forests expanded (mainly *Alnus* spp.) colonizing the open spaces previously dominated by hygrophilous plants during the Neolithic occupation (Figure 6). This expansion of alder carr environments on the shores of Lake Banyoles is consistent with data from SB2 sequence, where *Alnus* expanded from 5.5 kyr cal BP onwards (Figure 5) [38].

The importance of site formation processes and taphonomic issues in the interpretation of pollen records is shown in this study by the DCA in spatial analysis in both Sector A and Sector B-D (Figure 7) at La Draga. In both cases, the first Axis is explained by taphonomic processes. In Sector A, there is a contrast between layers (mainly UE2002) in the northern part, characterized by a high mineral content (determined by loss on ignition [49], low organic matter, and high values of pollen and spores resistant to erosion (*Pinus*, *Abies*, *Pteridium*, *Ophioglossum*-t); and, in the south, sediments characterized by high $CaCO_3$ content and high values of local-growing plants in edaphic soils, such as Asteraceae liguliflorae and *Typha-Sparganium*. Thus, while the first sediments were deposited through soil erosion episodes, the second indicate clear signs of edaphisation and authigenic deposition. In Sector B-D, there is an east-west gradient, showing a wet environment with accumulation of algae and hygrophilous vegetation to the east, and various fungal indicators of sub-aerial conditions to the west.

The comparative analysis of the pollen and NPP record in Sectors A and B-D was able to characterise the depositional environment of this new sector as well as demonstrate the important capacity of some non-pollen palynomorphs to define local environments. While the pollen record indicates similarities between Sectors A and B-D (Figure 6), NPP show a clear difference between the two sectors. Greater richness, ubiquity and abundance of fungi were documented in the waterlogged Sector B-D. While 47 new types were defined in Sector D (UAB types [15]), in Sector A no new types were defined and most of the ones documented in Sector B-D were not found in this emerged sector. This confirms the nature of fungal spores indicating humid and organic environments (*Diporotheca rhizopila*, *Sporoschisma* sp., *Valsaria* sp., *Diplocladiella scalaroides*, UAB-1, 14, 18A, 18B, 21, 32, 37, 40 (UAB-40 has been identified as *Xylomices*, a freshwater fungi [94]), 41B, 43, 44, 53), for those exclusively found in Sector D [49]). On the other hand, monolete spores, *Glomus*, *Sphaerodes*, Type 988 and types UAB-8, 22, 30A, 46, 47, 48, 50 are confirmed as indicators of mineral sediments deposited by soil erosion in terrestrial/sub-aerial environments. Thus, the NPP record confirms the deposition of mineral clays by soil erosion as the main process of sedimentation and the existence of a terrestrial/sub-aerial dry environment in Sector A.

Another remarkable feature is the scarcity of carbonicolous-lignicolous fungi in Sector A [93]. While the optimum conditions for development of these fungi (wood in decomposition and humidity) existed in Sector B-D, in Sector A decomposition of wooden elements would have occurred in a dry environment. This confirms the lignicolous nature of these fungi at La Draga, because if they were carbonicolous they should appear in higher frequencies (charcoal is very abundant in both sectors). On the other hand, coprophilous fungi are also scarce in Sector A (mean of 0.1%, maximum values of 1.8–2.6%), contrasting with high accumulations in the first phase in Sector B-D (mean of 1.9%, maximum values of 10–23%). Additionally, high values of parasite *Trichuris* sp. were documented in Sector D [48,95], while it was found in lower density in Sector A [49,96]. This scarcity of coprophilous fungi and parasites could be due to a lesser frequentation of domestic animals in this sector, or possibly due to taphonomic processes related to the nature of the depositional environment, as in the case of lignicolous fungi.

### 5.3.2. Social Use of Space

As stated in the introduction, palynological analysis of archaeological sediments (archaeopalynology) is a valuable tool to approach archaeological questions from a different perspective, given the influence of human activities in the pollen record composition. In this sense, significant accumulations of pollen grains and spores could respond to gathered plants, stored cultivars and presence of dung (animal and/orhuman) within the settlement. Despite taphonomic processes, in terms of deposition of pollen and spores, different environmental conditions played a major role in the spatial distribution of pollen and NPP at La Draga, while anthropogenic activities would also have influenced their distribution. They would have been responsible, by human use of space, for the existence of these different depositional environments, or the accumulation of certain taxa by anthropogenic activities or events.

The spatial analysis of pollen and NPP distribution in Sector B-D provided one example where the social use of space originated different depositional environments. While the space below a pile dwelling (east) [48] was characterised as humid and organic and waste material accumulated (faunal remains, coprophilous fungi, parasites), the exterior environment was defined as sub-aerial with higher values of arboreal pollen (AP). Thus, the differential deposition environments were caused by insolation in outsidespaces contrasting with moisture beneath the structures. Thus, the spatial distribution of AP can be used to assess inside/outside spaces in an archaeological site, in which the exterior areas provide a more reliable picture of the original pollen rain. On the contrary, inside spaces are strongly altered by human input of plants, giving a distorted image of pollen rain, with herbs overrepresented [97]. In that sense, some herbs (Poaceae, Asteraceae liguliflorae, Asteraceae tubuliflorae, Caryophyllaceae, *Polygonum*, *Aquilegia*-t, Apiaceae) accumulated in interior spaces in Sector B-D. These, may have been brought to the settlement for their subsistence value, as recorded in other archaeopalynological studies [98–100] and supported by the presence of most of these taxa in the carpological record at La Draga [30].

The identification of the human parasite *Trichuris trichiura* [95] points to the existence of human faeces at La Draga, concentrated in the NE part of Sector D [48], and suggests that spores of coprophilous fungi may not be restricted to animal origin. However, the occurrence of other parasites related to animals, such as swine (*Macracanthorhynchus* and *Ascaris* sp.) or ruminants such as cattle (*Dicrocoelium dendriticum* and *Paramphistomum* sp.) [95] indicates how humans and animals shared spaces within the settlement.

In addition, significant differences have been identified between some of the archaeological structures in Sector A, witha significant accumulation of anthropogenic taxa in structure E263, characterised by a large concentration of archaeological remains and an accumulation of bucrania and other ungulate skulls [101]. Figure 7 shows a positive correlation of Cerealia-t, *Aster*-t, Fabaceae, *Galium*-t, *Hypecoum*-t, *Verbena officinalis*, Cyperaceae, *Vitis*, and the accumulation of these taxa in the NE part of the sector, most likely related to the storage or processing of crops and gathering of some edible plants. In that sense, the spatial distribution of Cerealia-t pollen provided notable data about agricultural activities at La Draga. In Sector B-D, the highest values of Cerealia-t were documented in the south-western corner, spatially correlated with a concentration of grinding tools in the area nearby (Figure 9), as well as with charred monocot epidermis, signs of pollen degradation (maximum values of undeterminable pollen and lowest of pollen concentration) and a significant concentration of charred *Triticum aestivum/durum/turgidum* seeds, interpreted as a probable storage structure, indicating, consequently, a probable area of wheat processing [48]. In Sector A, Cerealia-t pollen is concentrated in structures E261 and E263 (Figure 9), also associated with a large accumulation of cereal grains in E263 (ongoing analyses), suggesting a probable crop processing or storage area.

### 5.3.3. Plant and Fungi Uses

Although other archaeobotanical remains, such as charcoal, seeds and fruits, provide more direct evidence of the use or consumption of certain plants, the occurrence of high

values of some short-dispersal pollen taxa in archaeological contexts can be interpreted as a result of human activities involving the use or consumption of plants. In that sense, a greater effort (in course) in relating these taxa to palaeoethnobotanical information on edible plants and other uses, and the correlation between the flowering season and when the consumed parts of plants are available, will provide valuable data about plant uses at La Draga, going beyond the well-known use of cereal crops and exploring the use of wild plants among farming societies.

In case of fungi, despite the existence of multiple species of edible fruiting bodies and the usefulness of some fungi for fire-lightning, some taphonomic and methodological issues must be considered in order to address the feasibility of their finding by palynological studies. In that sense, one factor which influences the NPP fossil record is the fact that only relatively big (heavy) fungal spores with thick walls are normally preserved. Most of the thin walled and small spores, which disperse better and which are known from the records of spores in the present atmosphere, do not fossilize [102] and have only been found occasionally in dental calculus [103]. In addition, the identification of spores at a taxonomic level is often difficult and, when identified, little is known about the processes of incorporation into the archaeological deposits and, as a consequence, their interpretation from an economic point of view is quite limited.

Edible fruiting bodies ("mushrooms") are Basidiomycota, belonging to the class Agaricomycetes, including many families (Agaricaceae, Boletaceae, Chantarellaceae, Russulaceae, among others) which produce small thin-walled hyaline spores which are not preserved in fossil pollen records. On the other hand, recent studies [93,104] showed how the gathering of bracket fungi was a frequent practice during the Early Neolithic occupation at La Draga. A total of 86 fruiting bodies were studied and six different taxa were identified. However, these taxa have not been identified among the fungal spores from archaeological sediments. Most tinder fungi documented at La Draga are Basidiomycota from the families Polyporaceae (*Coriolopsis gallica*, *Lenzites warnieri*, *Skeletocutis nivea*) and Fomitopsidaceae (*Daedalea quercina*); which also produce smooth hyaline spores that cannot easily be preserved in fossil records. In the case of another Basidiomycota, *Ganoderma adspersum* (Ganodermataceae) and an Ascomycota, *Daldinia concentrica* (Xylariaceae) the fungal spores have features that make them more resistant to decomposition and their invisibility among the fungal spores from soil samples should be explained as the result of factors other than post-depositional causes. In case of *Daldinia concentrica,* a high concentration of different spores belonging to Xylariaceae was identified in Sector B-D, but identification to species level is not possible given the absence of differential morphological traits. The absence of spores of *Ganoderma adspersum* (which can be preserved and identified) suggests that these bracket fungi were gathered, dried (preventing spore dissemination) and stored to be used by the inhabitants of La Draga.

## 6. Conclusions

This paper presents an overview on the potential of palynology within archaeoecological research to assess human-environment interactions and reconstruct past landscapes. The analysis of pollen and non-pollen palynomorphs in natural records was able to detect landscape transformation by Neolithic communities in the Western Mediterranean, specifically in Corsica and NE Iberia, providing proof of major land-cover changes induced by the first farmers, affecting sclerophyllous and riparian forests in North Corsica, Mediterranean maquis in South Corsica and oak forests in NE Iberia. The socioeconomic changes involved in the adoption of farming, including permanent settlement patterns, a growing population and more intensive productive activities increasingly reiterated in the territory, started a progressive process of landscape transformation, involving deforestation to open clearings for agriculture, grazing or permanent open-air settlements. Both in Corsica and NE Iberia, wetlands were attractive areas for these first farming communities, offering a variety of ecological niches which would have supplied various natural resources, with signs of human impact being detected from the Early and Middle Neolithic. In addition, palynology was

confirmed as a relevant source of data to address local palaeoenvironmental evolution in wetlands and in archaeological sites, providing evidence of flocks (spores of coprophilous fungi), and changes in hydrology (salinity, dryness/wetness, aquatic/palustrine phases) and in geomorphology (soil erosion indicators).

Finally, the analysis of pollen and non-pollen palynomorphs in archaeological sites (archaeopalynology) provided detailed information about site formation processes, social use of space and the use of plants and fungi in a lakeside settlement (La Draga, Girona, Spain). This work evidences the need to carry out complementary spatial and diachronic (profiles) analysis in archaeopalynological studies, in order to reconstruct palaeoenvironmental evolution in different phases of occupation and determine depositional and post-depositional events in the formation of the pollen record. The interest in studying the spatial distribution of pollen and NPP is based on the spatial heterogeneity of different taxa caused by human activities, in terms of inputs of plants to the settlement (gathering, cultivation, storage, etc.), and the arrangement of structures or space configuring different depositional environments or inducing soil erosion.

**Funding:** JR developed this research with a Juan de la Cierva-Formación contract (FJC2017) (MCINN, Spain), in the research group GAPS (2017 SGR 836). The InstitutCatalà de Paleoecologia Humana iEvolució Social (IPHES-CERCA) has been funded by MCINN in the programme of Units of Excellence 'Maria de Maeztu' (CEX2019-000945-M). The research in Corsica was possible thanks to the funding of the PCR "Approchegéoarchéologique des paysages de Corse àl'Holocène, entre mer etintérieur des terres. Trà Mare é Monti" and "Bouches de Bonifacio" research programmes, funded by the DRAC Corsica and was part of the MISTRALS-PALEOMEX programme of CNRS (INEE-INSU scientific departments). Theresearch in NE Iberia waspossiblethankstotheproject 'Paleoambiente, modelización del paisaje y análisis del uso de plantas en la transición a la agricultura en el Noreste de La Península Ibérica (PID2019-109254GB-C21)' fundedby MCINN (Spain).

**Institutional Review Board Statement:** Not applicable.

**Informed Consent Statement:** Not applicable.

**Data Availability Statement:** Following the ORDP, and aiming to improve and maximize the access to the research data, pollen datasets from natural deposits presented in this work have been submitted to the European Pollen Database (EPD, http://www.europeanpollendatabase.net/) (accessed on 27 September 2017 (SB2 Banyoles), 4 February 2020 (Saint Florent) and 15 April 2020 (Piantarella)).

**Conflicts of Interest:** The author declares no conflict of interest.

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
