# Peer review of "The Role of Palynology in Archaeoecological Research: Reconstructing Human-Environment Interactions during Neolithic in the Western Mediterranean"

_applsci, doi:10.3390/app11094073_

Round 1

Reviewer 1 Report

Palynological analyses are essential for understanding of the Late Cenozoic terrestrial evolution. The reviewed manuscript is a proof. Taking into account new data from some localities of Corsica and NE Spain, the author reveals the Holocene vegetation changes and the relevant human-environment interactions in the Western Mediterranean. The paper is informative and contributes significantly to the subject. My main concern is the manner of writing, which I suggest to improve greatly (see below). My recommendations are specified below.

  • To me, this paper resembles more a review than a research article. So, I propose to re-label it as Review.
  • Abstract: I kindly ask you to be more consistent in your abstract – please, re-read it and compose in more logical way.
  • Key words: please, avoid words already used in the title.
  • Section 2: you need to re-organize this section. It should consist of two subsections devoted to Corsica and NE Spain. In each, you need to clarify the modern geographical setting and the archaeological knowledge of the region. If you wish also to tell about the broad context of Western Mediterranean, you need to add a section where this context is explained for both modern geography and archaeology.
  • Why pollen percentage diagrams of La Draga differs from three others?
  • Line 537: what do you mean under 'natural deposits'?
  • Conclusions: can you provide a numbered list of 5-7 main findings?
  • The writing is generally appropriate, but it needs everywhere (!) a) certain linguistic polishing, b) bigger consistency (for instance, if you start with Corsica and then turn to NE Spain, this order should be followed everywhere).
  • I recommend the author to make less emphasis on the importance of palynological methods in archaeology (this is undisputable and widely-known) and to focus more on clear and consistent description of his findings. This is necessary to do in the entire text.

Please, do not find me overcritical. I really like this paper, and I just wish to see it as good in regard to the manner of writing, as it really deserves.

Author Response

Dear,

Thank you for your comments and suggestions, which helped to improve the manuscript. Glad to know you find it interesting. Below you will find my replies:

1- To me, this paper resembles more a review than a research article. So, I propose to re-label it as Review.

I think it is probably a good suggestion, but this should be decided by the editors… If editors think that this paper should be re-labelled to Review I will agree them.

2- I kindly ask you to be more consistent in your abstract – please, re-read it and compose in more logical way.

The abstract was reviewed, avoiding repetition and has been composed in a more structured way.

3- Key words: please, avoid words already used in the title.

The keywords ‘pollen analysis’, ‘Western Mediterranean’ and ‘Neolithic’ were removed, and replaced by ‘bioarchaeology’, ‘Iberian peninsula’, ‘Middle Holocene’

4- Section 2: you need to re-organize this section. It should consist of two subsections devoted to Corsica and NE Spain. In each, you need to clarify the modern geographical setting and the archaeological knowledge of the region. If you wish also to tell about the broad context of Western Mediterranean, you need to add a section where this context is explained for both modern geography and archaeology.

I re-organized Section 2, now mentioned ‘Study area: environmental settings and archaeological background’, and divided in 2 sub-sections: 2.1. Corsica and 2.2. NE Iberia. The paragraph about Neolithisation in the Mediterranean was placed in the introduction.

5- Why pollen percentage diagrams of La Draga differs from three others?

Because it consists of a pollen diagram from an archaeological site, and the others from natural (lacustrine-wetlands) deposits. In archaeopalynological diagrams it is better to use bars rather than continuous silhouettes to represent the percentages of different taxa, because there is not a direct temporal continuity between layers as it occurs in natural deposits.

6- Line 537: what do you mean under 'natural deposits'?

It’s true that the use of ‘natural deposits’ in this sub-section title was confusing. I changed it by ‘lakes and wetlands’.

7- Conclusions: can you provide a numbered list of 5-7 main findings?

The main findings are clearly described in the discussions. In the conclusions section I preferred to sum up, shortly, the main outcomes of this paper, one paragraph about landscape transformation detected in natural deposits (lake, wetlands); one paragraph about the application of pollen analysis in archaeological sites (archaeopalynology). I think this is a question of style, I express my gratitude to the reviewer suggestion, but I prefer to write the conclusions in this way.

8- The writing is generally appropriate, but it needs everywhere (!) a) certain linguistic polishing, b) bigger consistency (for instance, if you start with Corsica and then turn to NE Spain, this order should be followed everywhere).

About certain linguistic polishing… This paper was reviewed by a native English expert in revision of scientific manuscripts in the field of archaeology and palaeoecology (Mr. Peter Smith).

I checked what the reviewer suggested about bigger consistency. I followed always the same order. In sections 2, 3, 4 and 5, firstly, I talked about South Corsica, then North Corsica, NE Iberia and La Draga.

9- I recommend the author to make less emphasis on the importance of palynological methods in archaeology (this is undisputable and widely-known) and to focus more on clear and consistent description of his findings. This is necessary to do in the entire text.

I find very necessary to make emphasis on the importance of the application of palynological methods in archaeology, as, at least in my study area, there are many excavations where sampling for pollen analysis is not considered in their research. This is one of the main aims of this paper, to make people concern about the potential of archaeopalynology. I think the findings are clearly described and the emphasis on the importance of palynological methods is complementary and does not reduce the clarity of the manuscript.

Reviewer 2 Report

Please excuse the delay of my review. I had an accident and my middle finger is now a little bit shorter. So much for explanations, here is the review.

This is one of the best manuscripts I have seen in a long time and a delight to read. It is well constructed with very few script errors. Those few that there are can be dealt with at proof stage, but are listed below. The arguments are sound and well-considered and the outcomes eminently sensible. The analyses appear to be well done and I wish all the papers I have to review were as good as this one. I recommend acceptance without revision. My compliments to the author.   With best regards  15th April 2021     Consistency hyphen and en-dash:   Please check the manuscript for consistent use of hyphen and en-dash.

examples below:

line 116:    8.5-8.2 kyr cal BP

line 328: ~5.9–5.8 and ~5.5 kyr cal 

  Italics for genera and species names:   All species and genera names in the figure text for Figure 3, Figure 5, and Figure 6   Line 292: Pinus Line 295: Erica Line 323: Alnus Line 324: Pinus, Erica Line 344: Quercus Line 345: Corylus Line 348: Quercus  Line 388: Quercus, Corylus, Fraxinus Salix Line 389: Ulmus, Alnus Line 393: Quercus  Line 473: Erica  

Author Response

Dear,

Thank you for your positive comments. Glad to see you find interesting my contribution to this SI in Applied Sciences. I checked the use of hyphen and en-dash in the whole manuscript. About the following comment:

2-  Italics for genera and species names:   All species and genera names in the figure text for Figure 3, Figure 5, and Figure 6   Line 292: Pinus Line 295: Erica Line 323: Alnus Line 324: Pinus, Erica Line 344: Quercus Line 345: Corylus Line 348: Quercus  Line 388: Quercus, Corylus, Fraxinus Salix Line 389: Ulmus, Alnus Line 393: Quercus  Line 473: Erica  

It’s true that in the pdf version these genera and species names are not in italics, but in the word file they are correctly, so I did not make any change.